# The use of GRDC gauging stations for calibrating large-scale hydrological models

Peter Burek,[1] Mikhail Smilovic[1]

[1] International Institute for Applied Systems Analysis, Laxenburg, Austria

*Correspondence to*: Peter Burek (burek@iiasa.ac.at)

**Abstract.** The Global Runoff Data Centre provides time series of observed discharges and information on hydrometric stations that are valuable for calibrating and validating the results of hydrological models. We address a common issue in large-scale hydrology that has not been satisfactorily solved, though investigated several times. To compare simulated and observed discharge, grid-based hydrological models must fit reported station locations to the resolution-dependent gridded river network. We introduce an Intersection over Union ratio approach to selected station locations on a coarser grid-scale, reducing the errors in assigning stations to the correct upstream basin. We update the 10-year-old database of watershed boundaries with additional stations based on a high-resolution (3 arc seconds) river network and provide source codes and high- and low-resolution watershed boundaries to easily select stations for calibration/validation of hydrological models. The dataset is stored on Zenodo with the associated DOI https://doi.org/ 10.5281/zenodo.6906577.

## 1 Introduction

River discharge is one of the most important variables in hydrological modeling because all basin processes are integrated into this variable. Discharge spatially and temporally integrates the range of meteorological variables and basin characteristics. Spatially and temporally distributed properties of river and lake morphology, soil, groundwater, snow, glaciers, climate, land cover, and human interaction influence discharge at the outlet of a basin. Discharge is extremely useful for calibrating and validating hydrological models using different objective functions, such as Nash-Sutcliffe (Nash and Sutcliffe, 1970) and Kling-Gupta (Kling et al., 2012) for calibrating global hydrological models (Müller Schmied et al., 2021; Sutanudjaja et al., 2014; Hanasaki et al., 2008). It is also useful for tasks like estimating flood hazards (Alfieri et al., 2015), inland navigation (Nilson et al., 2013; Christodoulou et al., 2020), energy power production (Hunt et al., 2020; Van Vliet et al., 2016), and water scarcity (Hoekstra et al., 2012; Van Beek et al., 2011).

Since the establishment of the Global Runoff Data Centre (GRDC) database (Vorosmarty et al., 1998; Fekete and Vörösmarty, 1999), the number of stations has increased, and the number of publications using the GRDC dataset is also growing - the GRDC publication database of 2021 (GRDC, 2020) references 118 publications using the discharge time series. The Generic Statistical Information Model (GSIM) database (Do et al., 2018) provides a good overview of several river discharge databases

worldwide. Although there are several public databases of river discharge at a basin-scale (e.g., Mekong basin, Mekong River
      Commission (2020)), the GRDC database offers the richest source of global river discharge data, as follows:

> "A global hydrological database is essential for research and application-oriented hydrological and
> climatological studies at global, regional, and basin scales. The Global Runoff Database is a unique
> collection of river discharge data on a global scale. It contains time series of daily and monthly river
discharge data of well over 10,000 stations worldwide. This adds up to around 470,000 station-years with
> an average record length of 45 years" (GRDC, 2020).

      The GRDC database of river discharge comes which information about the stations from the data providers, like the location
      of the station, name of the station and the river, upstream area, elevation, mean discharge, and more. Especially the location
and the upstream area are very important to compare model results from hydrological models with station discharge data.
      Quality checking of station attributes and spatial redistribution of station locations for different gridded river networks for
      hydrological models have been carried out since the beginning of GRDC data collection (Fekete and Vörösmarty, 1999) and
      for each model again and again (Sutanudjaja et al., 2018; Zhao et al., 2017; Wang et al., 2018). For example, to test the
      Community Water Model (CWatM) (Burek et al., 2020) global model performance on 30 arc minutes (=0.5° ~ 50 km x 50
km, hereafter, 30'), we used the station data and the global drainage direction map (DDM30) network (Döll and Lehner, 2002)
      and corrected the locations to fit with the approach of Zhao et al. (2017). Several errors can occur when the discharge station
      is used for gridded hydrological modeling, as follows:

a)    The station location is not at the correct location and is too far from the river.

b)    The station location is at the correct location, but because of the river width and/or the grid resolution of a high-
resolution river network, the station location is not in the suitable grid cell of the river network or because, even
      at 100 meters, the network is not high-resolution enough to capture the station location.

c)    The high-resolution network does not represent reality (e.g., the river does not flow in the deepest part of the
      valley because of human interventions).

d)    Upscaling error. When a coarser resolution for hydrological modeling is applied (e.g., 30'), using the original
station location might lead to its position being wrongly assigned because, for instance, the coarser grid cell river
      network may include the junction with a tributary. In contrast, the station may indicate the tributary itself.

e)    Mismatch error. Suppose the station location is selected only by comparing the upstream area of the upscaled
      network with the reported upstream area. In that case, a station could be assigned to the wrong basin because the
      upstream area fits slightly better.

f)    Global station density is unevenly distributed. We find a high density for North America and Europe and a low
      density for Asia and Africa. In North America or Europe, some stations are close downstream to other ones,
      even though no significant tributaries are entering.

This paper aims to provide a Python code to easily select stations for calibration/validation of hydrological models by addressing these possible errors and giving examples of how to correct them. Lehner (2012) has already calculated explicit watershed boundaries for 7,163 basins on a high-resolution network. These watershed boundaries are freely available on the GRDC webpage (GRDC, 2020). We repeated this exercise, but using a higher-resolution network based on a more up-to-date dataset, the 3 arcseconds (~100m or exactly 92.61 m at the equator, hereafter, 3'') MERIT hydro-network (Yamazaki et al., 2019) rather than the 15 arcseconds (~500 m) HydroSHEDS (Lehner et al., 2008). Moreover, we used a greater number of GRDC stations added in the last ten years (10,701 stations as opposed to 7,532). In addition to the high-resolution basins, we added a method for upscaling each station automatically to 5' (~10 km) and 30' (~50 km) using a more advanced method than simply comparing the river network upstream area with the reported upstream area. Using this method, a selection of stations can be appropriated to the resolution of the hydrological model. Furthermore, our code is available and open source in Python to change or calculate stations for individual applications.

## 2 Methods

The methods can be split up into three main groups, each group building upon the results of the previous one. The first method describes allocating a station location from the GRDC database to fit best on a high-resolution network. This method reproduces the approach from Lehner (2012). The second method describes how to upscale the station location from a high-resolution network to a low-resolution network used in standard land-surface hydrological routing models by comparing upstream area and similarity of the station upstream areas in high and low resolution. The third method describes how to select the most appropriate stations for calibrating hydrological models, depending on the metadata of the stations and the chosen model grid resolution.

### 2.1 Procedure for station allocation on a high-resolution network

### 2.1.1 Automatic procedure

We used the MERIT hydro database of Yamazaki et al. (2019), which comes as an open-source database in chunks of 5° * 5° at 3'' resolution (36 billion grid cells per 5° * 5°). We used the river network direction maps and applied the D8 flow model convention: either each grid cell can flow into one of the eight neighboring grid cells, or it is a sink. This approach, which does not allow rivers to be split into two streams or grid cells to contribute to several basins simultaneously, is used in most Land-Surface Models and grid-based hydrological models. We obtained the upstream area of each high-resolution grid cell from the upstream area in $km^2$ from the MERIT dataset. For the evaluation, we used all stations with a reported upstream area greater than or equal to 10 $km^2$ (124 stations have an upstream area smaller than 10 $km^2$) or with no reported upstream area record (327 have no upstream area record in the GRDC dataset).

For the automatic station allocation, we mainly follow the protocol of Lehner (2012):

1. A rectangular search radius of 165 arcsec (~5 km) for each station was defined.

2. For each grid in this rectangle, the upstream drainage area (UPA) from the network from Yamazaki et al. (2019) was compared to the area reported in the GRDC, and the upstream area accordance is computed:

        Upstream area accordance = GRDC reported UPA / gridded network UPA

(where: GRDC reported UPA < gridded network UPA)

        Upstream area accordance = gridded network UPA / GRDC reported UPA

        (where: GRDC reported UPA ≥ gridded network UPA)

3. All cells with an upstream area accordance of less than 50% were dismissed from further evaluation.

4. A first ranking scheme – area discrepancy (RA) - was calculated with values between 0 (best fit) to 50:

RA = 100 - Upstream area accordance[%]

5. For the second ranking scheme – distance (RD) – the distance of the cell to the reported station location in the GRDC database was calculated and normalized to get the value 0 at the station location and 50 in 5 km distance.

6. An objective criterion (OC) for ranking was computed by OC = RA + 2 * RD. The equation and weighting were taken from Lehner (2012).

7. The grid cell with the lowest OC value was taken as the corresponding grid cell for the station location on a high-resolution network

8. If no station location was found in this step, the search radius was increased to 5' (~10 km), OC was calculated as OC = RA + RD, and the lowest OC value was taken as the corresponding grid cell.

### 2.1.2 Manual procedure for the remaining stations

For the ~7.5% of the stations that failed both rounds of searching, we carried out the following manual inspections:

- 3% of the stations in the GRDC database (327 stations) have a reported area of -999. For these stations, we used the next biggest river. We checked manually with GIS, but we did not check all station information manually (e.g., station name, river name, and altitude).

- 1.5% of stations that failed the automatic search but had a valid UPArecord (169 stations), we manually checked and

assigned a location in the range of up to 120 km from the original site (to address any typographical error, e.g., 51.57° instead of 52.57°)

- For 2.2% of the stations (228 stations), we could not find an adequate location on the high-resolution network - due perhaps to errors in the GRDC database or insufficient network maps (e.g., missing canals, braided rivers, diversion, and confluent rivers)

### 2.1.3 Output: Polygons of basins

For 10,349 basins, we assigned polygons based on the reallocated station locations on the high resolution (3'') with the Python library flwdir from Eilander et al. (2021) and the river direction mosaic maps from Yamazaki et al. (2019). Like the original from Lehner (2012), we produced two versions: a) polygons that follow the exact grid cell contours with high memory requirements, and b) a version with smoothed edges and low memory consumption. The resulting shapefiles were produced in the ESRI shapefile format and included the station information from GRDC. This process addresses errors a), b), and c) (noted above) and provides an update to the shapefiles of Lehner (2012)

### 2.2 Upscaling station location to a coarser grid cell resolution

The main idea of creating a new set of high-resolution watershed boundaries was not to update the work of Lehner (2012) but to use a different method of assigning station discharge time series to the correct grid cell in grid-based hydrological models. Global hydrological models use 30' resolution in the ISIMIP3 project (Warszawski et al., 2014). The trend for global modeling is to move toward higher resolutions at 5' and hyper resolution ($\leq$ 1km) (Bierkens, 2015). For regional studies (Hanasaki et al., 2022; Guillaumot et al., 2022), the resolution is already 1 km or below. Approaches to upscaling to coarser resolutions are mainly based on comparing reported UPA and UPA calculated from the river network (Fekete and Vörösmarty, 1999; Zhao et al., 2017).

With the flwdir tool from Eilander et al. (2021) and with the idea of Munier and Decharme (2021) of comparing the similarity of shapes, it is possible to introduce another objective criterion - the similarity of high-resolution watershed boundaries and low-resolution boundaries. Using the method of comparing upstream areas, we can partly address error (d) upscaling error but not errors (e) and (f).

For an automatic upscaling process, we followed this protocol:

- We defined a minimum UPA for the station we wanted to use in the low-resolution hydrological model (e.g., UPA $\geq$ 9,000 km$^2$ for 30' (~3 cells), UPA $\geq$ 1,000 km$^2$ for 5' (~12 cells)).
- To find the grid cell on the coarse resolution network which fits best to the UPA and shape of the high-resolution network, we calculated two objective criteria for all coarse grid cells with a distance <= 2 coarse cell distance (altogether 25 grid cells) to the location of the station on the high-resolution network (see figure 1)
- For each coarse grid cell, the coarse UPA was derived, and the upstream area accordance was computed as first objective criterion.

upstream area accordance = GRDC reported UPA / coarse UPA (GRDC reported UPA < coarse UPA)

upstream area accordance = coarse UPA / GRDC reported UPA (GRDC reported UPA $\geq$ coarse UPA)

The upstream area accordance can have a value between ]0,1], with 1 having GRDC and coarse UPA having the same value.

- The second objective criterion was the Intersection over Union ratio (Rezatofighi et al., 2019; Munier and Decharme, 2021). Intersection over Union ratio = Area of intersection / Area of union

Therefore the watershed shape for the station on high-resolution and the shape for each coarse grid cell was created. The area of intersection represents the area the high-resolution and the low-resolution shape have in common. The area of union represents the area of the combined shapes of high and low resolution. The Intersection over Union

ratio can have a value between [0,1]. The closer to 1 the Intersection over Union ratio value, the more similar the shapes are.

- To reduce the two objective criteria to one solution, the minimum Euclidian distance between the best possible solution (at 0,0) and the two objective criteria was calculated. Both objective criteria have a range between 0 and 1. Therefore, we decided to use a weighting factor of 1 for both criteria.

$$ED = \sqrt{(1 - Upstream\ area\ accordance)^2 + (1 - \text{Intersection over Union ratio})^2}$$

- The coordinates on low-resolution with minimum Euclidian distance were chosen as the station coordinates for this grid size resolution

Figure 1 illustrates this method for low resolution 5' and for cell location No. 7, which is one 5' cell south of the cell

where the station "Passau/Inn" is located (see the zoom in the upper left part of figure 1). Even if this cell is not representing the cell where the station is located, this cell fits the upstream area accordance and the Intersection over Union ratio best of all 25 cells around the station location.

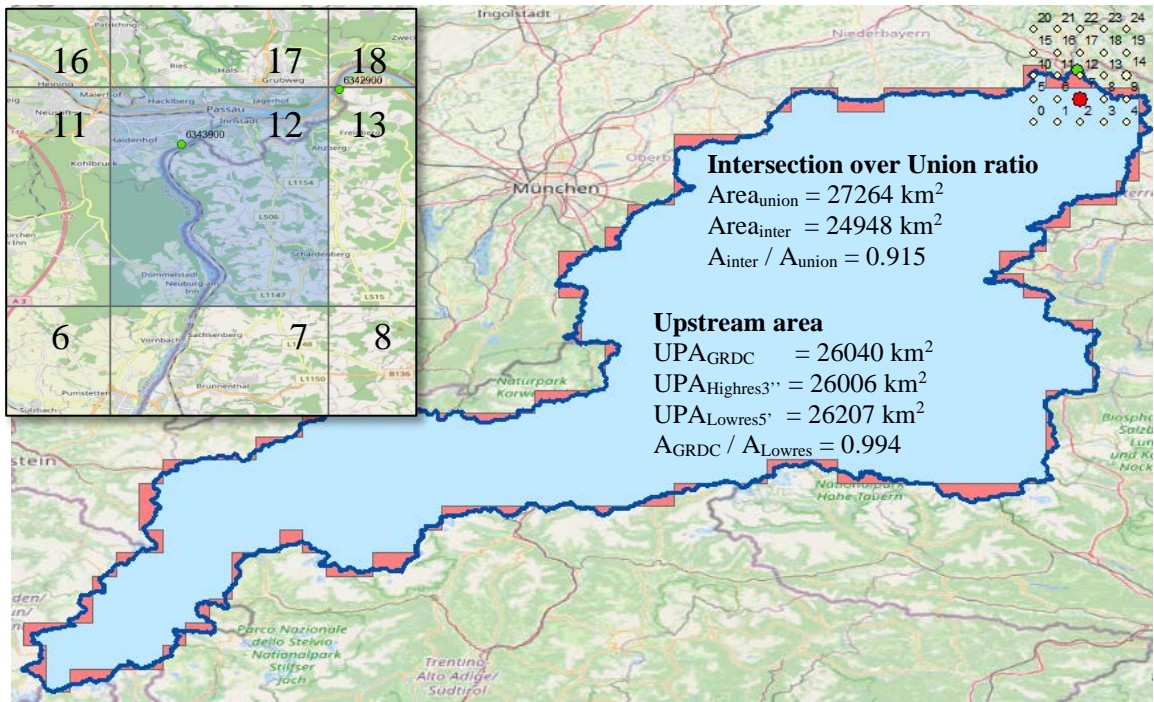

**Figure 1:** Intersection over Union ratio for the River Inn basin at Passau, Germany, GRDC 6343900. The dark blue line is the watershed of the Inn to Passau at a 3" high resolution. The light blue is the intersection between the low-resolution Inn at 5' and 3'', and red signifies the union of the 5' basin with the 3'' basin. **).** © OpenStreetMap contributors 2022. Distributed under the Open Data Commons Open Database License (ODbL) v1.0.

Figure 2 shows four examples of the 25 cell locations around station "Passau/Inn". Figure 2a uses the cell where the station is located. This cell represents not only the Inn, but the also the Danube and the Inn basin. Figure 2b includes only a small tributary of the Inn and figure 2c contains only the Danube basin but not the Inn basin. Figure 2d shows the best location (one grid cell south of the grid cell with the station – same as in figure 1).

As a result, we obtain a pair of coordinates for each station on a coarser resolution. Here we chose 5' and 30'. For 5', the network from Eilander et al. (2021) was used based on the high-resolution network from Yamazaki et al. (2019), while for 30', the DDM30 network from Döll and Lehner (2002) was used, as this is the agreed network for the ISIMIP2 and ISIMIP3 (Frieler et al., 2016) hydrological modeling effort.

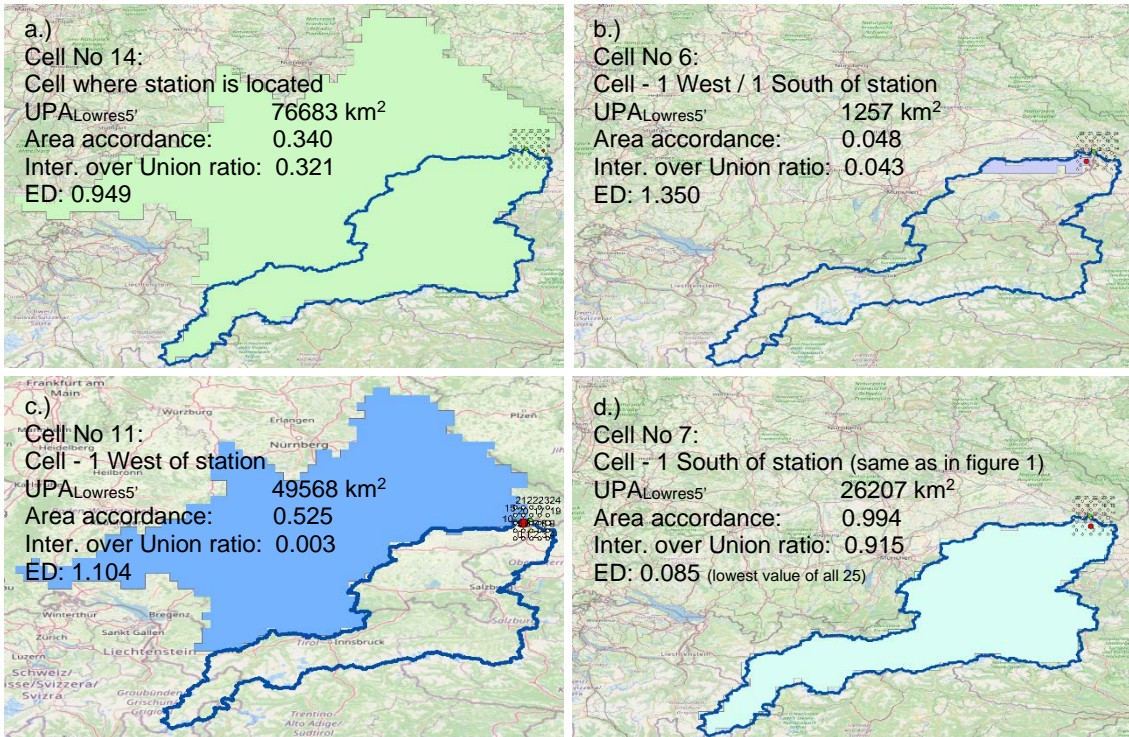

**Figure 2:** Concept of similarity for the station Passau, Inn, Germany - GRDC 6343900 with a high-resolution watershed map shown as a blue outline and four different watershed maps based on a 5' resolution network around the station location. © OpenStreetMap contributors 2022. Distributed under the Open Data Commons Open Database License (ODbL) v1.0.

### 2.3 Selecting station for calibration of a hydrological model based on metadata

In the previous step, we selected stations based solely on location metadata. For the next selection step, we included meta information of time series like length, end date, and the number of missing values in a time series. For calibration or validation, the unsuitable stations were those with only short time series, those that ended too far in the past, and those with too many missing values. The criteria for "too short" and "too old" are subjective and can be chosen in another way, as in Alfieri et al. (2020), but if the criteria are not strong enough, a post-selection can be done. If they are too rigid, the settings part of the Python code can be changed. Fortunately, all the necessary information is included in the metadata file from GRDC.

#### 2.3.1 Deselection and ranking criteria

We included several criteria for selecting or deselecting a station (see table 1). We derived the first two criteria from the previous analysis of the station location.

- The accordance of the UPA on the chosen resolution with the area reported from GRDC: here, we chose a relatively forgiving upstream area accordance. If the upstream area of the selected resolution had a criterion of more than 40%, this criterion is fulfilled. In most cases, the area was above this ratio, but we did not want to deselect stations where the GRDC record might be accurate.

- The Intersection over Union ratio between the high-resolution shapefiles and the shapefile of the chosen resolution: the high-resolution shapefile was built on the 3'' MERIT network data. The 30' had a different source and went through an upscaling approach. Therefore, more significant discrepancies between the high-res shapefile and the low-res 30' were possible.

We included two selection criteria from the metadata information about the time series.
- The time series should have at least five years of monthly or daily records.
- The end date of a time series should be later than 1985.

**Table 1:** Selection criteria based on low-resolution

| Name of criterion | Selected at 30' | Selected at 5' |
|---|---|---|
| Intersection over Union ratio | 70% | 80% |
| Ups. area accordance | 40% | 40% |
| Years in time series | 5 years | 5 years |
| End date | 1985 | 1985 |

## 2.3.2 Division of stations

The stations may be too close to each other for it to be worth calibrating both. We checked the similarity of the low-res shapefiles. If they were equal to or more than 95% similar, we decided to calibrate only one station and keep the other for validation purposes. To choose which of the similar stations we kept for calibration, we introduced a ranking/scoring system. If a station had a higher Intersection over Union ratio or upstream area accordance than 80% it got one scoring point for every 2%. Stations earn scoring points for every five additional years of time series length and for end dates of the time series after 1985. For missing data in the time series, scoring points are subtracted (see Table 2 for the scoring criteria). The station with the higher scoring points is chosen. These criteria are subjective and can be changed in the Python code.

**Table 2:** Scoring where two stations are too similar

| Name of scoring | 0 points at | 1 point for every | Max/Min points |
|---|---|---|---|
| Intersection over Union ratio | 80% | 2% | 10 |
| Ups. area accordance | 80% | 2% | 10 |
| Years of time series | 5 years | 5 years | 10 |
| End date | 1985 | 3 years | 12 |
| Missing % | 100% | Neg. point for 5% | -20 |

### 2.3.3 Output: List of stations to be appropriated for calibration

As a result of this step, we obtained two tables for 30' and 5' that distinguished the stations as useful for calibration, stations that could be used for validation, and stations that were not recommended for calibration or validation. All evaluation was solely based on the metadata file provided by GRDC.

## 3. Results

### 3.1 Station allocation on high resolution

The March 2022 GRDC station dataset has 10,701 stations in total. We used only those stations with an UPA equal to 10 km$^2$ (thus discounting 124 stations), but we kept the stations without data for the UPA (327 stations). Using automatic detection of the most appropriate high-resolution MERIT network on 3'' and with manual search (for 169 stations), we still had 228 stations we could not assign to a basin. Figure 3 shows the global distribution of GRDC stations (status: March 2022), with a high concentration of stations in North America and Europe and a lower and more clustered distribution in Africa and Asia.

For further analysis, we had 10,349 stations with a counterpart in a location on the MERIT network and an assigned basin on 3'', and 49 of these basins did not have a reported area in GRDC. From the remaining 10,300 stations, we calculated the area in accordance with the GRDC reported UPA and the area calculated on the high-resolution network using UPA maps from Yamazaki et al. (2019). We kept only those with an upstream area accordance ≥ 0.4 (1,0241 stations). For hydrological modeling, this accordance might be too low. Still, we assumed there were some errors in the reported area of GRDC and that these stations could be deselected in a further step, if necessary.

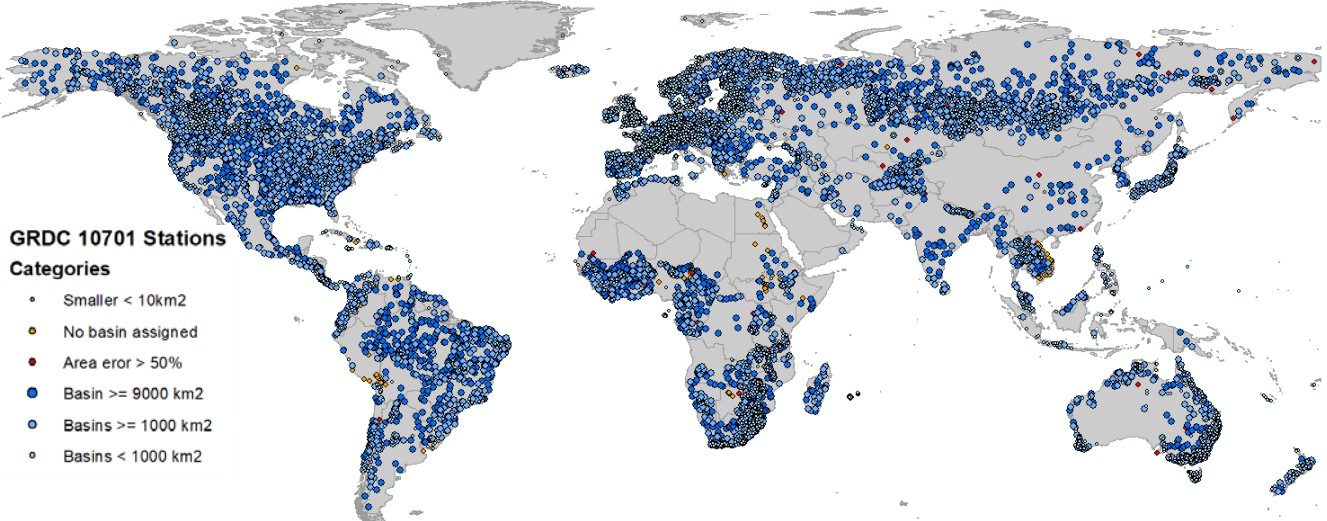

**Figure 3:** Location and categories of 10701 GRDC stations (World administrative boundaries by https://www.opendatasoft.com)

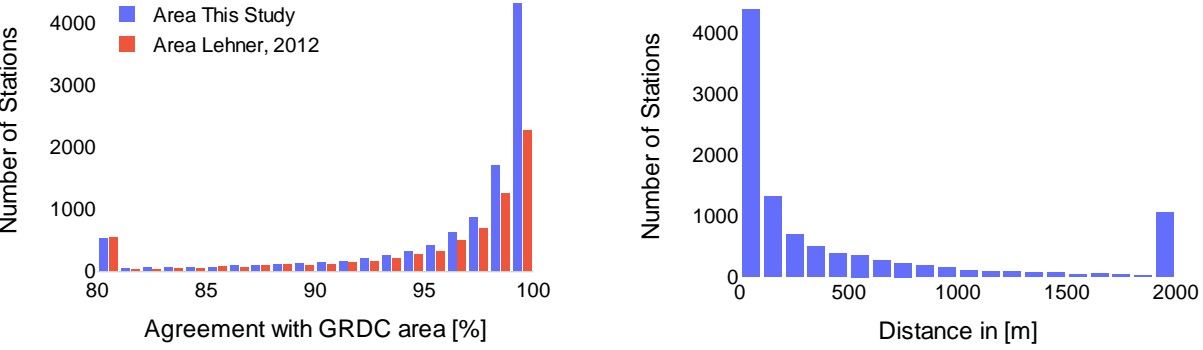

**Figure 4:** Histogram of (a) accordance with GRDC upstream area and (b) distance from the corrected location to the GRDC location

The histogram of upstream area accordance in Figure 4a and Table 1 shows that a high number of stations (43%) have upstream area accordance that is equal to or more than 99% (an area error of less than only 1%). Compared with the work from Lehner (2012), we obtained a slightly higher percentage of accordance in this class but almost twice as many stations (4,332 vs. 2,422).

88% (85% for Lehner (2012) ) still had good accordance of 90% or more. 330 stations had an area accordance of less than 75%, 18 stations of less than 50%, while 49 stations had no area reported.

Figure 4b shows the distance in meters from the reported station coordinates in GRDC and the station location according to the high-resolution network. A necessary shift in stations might be required because a) the river network is not high-resolution enough to capture the river (see Figure 5d), or the river width is greater than 90 meters, and it would be necessary to shift the

station location from the river shore into the middle of the river to match with the high-resolution network. However, we assumed that most distance errors greater than 500 meters come from the stations being wrongly allocated. Table 3 shows that the percentage of upstream area accordance negatively correlates with the distance median. (Here, the distance is the distance in meters between the reported station location in the GRDC dataset and the location represented in the 3'' MERRIT network. The median of distance is calculated as the median of distances for all stations in each row of table 3.)

**Table 3:** Comparison of stations with accordance in area of high-resolution basins with area reported from GRDC

| Percentage of Area Accordance | Number of Stations This Study | % This study | Median of Distance [m] | Number of Stations Lehner (2012) | % Lehner (2012) | * of 7163 station in Lehner (2012), 7025 also match with the new dataset |
|---|---|---|---|---|---|---|
| ≥ 99 | 4332 | 42 | 98 | 2422 | 35 | |
| ≥ 95 | 7920 | 77 | 180 | 5043 | 72 | |
| ≥ 90 | 8980 | 87 | 203 | 5888 | 85 | |
| ≥ 85 | 9446 | 92 | 382 | 6287 | 90 | |
| ≥ 75 | 9862 | 96 | 418 | 6627 | 95 | |
| ≥ 50 | 10174 | 99 | 661 | 6922 | 99 | |
| ≥ 0 | 10300 | 100 | 1306 | 6976 | 100 | |
| no area | 10349 | | | 7025* | | |

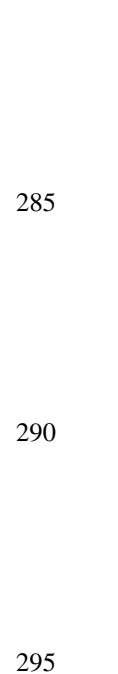

**Figure 5:** Possible errors in station location. © OpenStreetMap contributors 2022. Distributed under the Open Data Commons Open Database License (ODbL) v1.0.

Figure 5 shows some of the possible errors as well as the need to correct station location that will be used in hydrological models:

Part a) GRDC station 1643220 Mouila-Val-Marie, Ghana, has a reported UPA of 15,900 km$^2$. The closest river to the station location has a high-resolution UPA of 2,477 km$^2$. The next location with a closer upstream area to the reported one (15,868 km$^2$) is 50 km to the west. Thus, here, either the station location or the reported area is wrong.

Part b.) GRDC station 1737300 Bamingui, Central African Republic, with a reported area of 4,380 km$^2$, has no river of this size at a closer distance. We thus chose the river near the city of Bamingui with a high-resolution UPA of 6,075 km$^2$ at a distance of 25.7 km from the GRDC station location.

Part c) shows the station location of GRDC 1834101 Lokoja, Niger. The station seems to be on the River Benué, a tributary of the Niger River. According to Udo et al. (2021), the station is located on the Niger River, upstream of the junction of the Niger and Benué. Lehner (2012) assumed the station is downstream of the junction. No area is reported as being associated with this station, but the area could be 337,000 km$^2$ (Benué), 1651,000 km$^2$ (upstream Niger), or 1990,000 km$^2$ (downstream Niger).

Part d) shows the GRDC station 1837450 at Challawa Bridge in Nigeria at the exact location underneath a bridge over the

315 River Challawa. However, the high-resolution network of 90 meters shows no river on this grid cell, and the station location must thus be shifted 90 meters to the west.

Another example is GRDC stations 1396200, 1396201, and 1396210, which have the same reported location but different river station names and UPAs.

For GRDC station 4208919 on the Dunkirk River, Canada, we found a typographical error. Instead of 58N, it should be 56N. Station Siramakana, Mali, at the river Baoule GRDC station 1112330 is, according to Hydroscience_Montpellier (2022), around 50 km from the station location mentioned in the GRDC dataset.

The remaining 10,241 stations are not equally distributed globally. There are regions where water cannot be measured as

streamflow, such as Greenland, the Sahara, the Arabic peninsula, the Kalahari, and Central Australia. In other regions, we know that streamflow is measured, but GRDC does not have the records (some parts of Italy, Indonesia), and some regions where we do not even know if there are measurements (e.g., North Korea). Some basins, especially in North America and Europe, have a dense reported discharge station network (e.g., the Danube). Figure 6 shows the number of subbasins of GRDC stations placed inside the other. For example, in the Danube, there is a station for the upper River Inn, which is inside the basin

of the lower Inn (another station), which is inside the basin of the upper Danube, which is inside the basin of the middle Danube, just like the concept of stacked dolls (Matryoshka dolls).

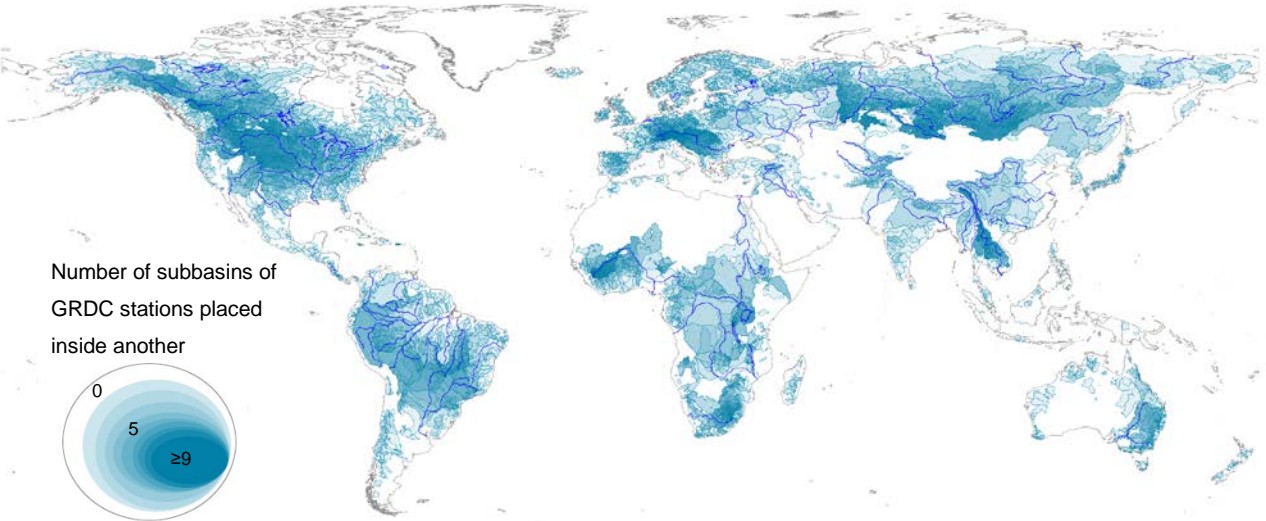

**Figure 6.** Watershed shapefile of 10241 station using GRDC stations and MERIT network map
(World administrative boundaries by https://www.opendatasoft.com)

### 3.3 Station allocation on low resolution 5' and 30'

We allocated 10,241 stations with an area $\geq 10 \text{km}^2$, and after creating shapefiles for each station, we created shapefiles and a station record for low-resolution of 30' and 5'. For the resolution of 30', we used a threshold of an UPA of $\geq 9,000 \text{ km}^2$ (around three grid cells on 30'). For 5', we used a threshold of $\geq 1,000 \text{km}^2$ (~12 grid cells on 5'). This selection was subjective, and 340 other papers have slightly different assumptions (Alfieri et al., 2020).

With the similarity method, we can avoid that a basin being allocated to a station that fits better by the UPA but is not very similar to the basin shapefile at high resolution. Figure 7 shows this for two examples. The station "Above Babine River" of the Skeena River in Canada, GRDC No. 4245920, is the station shortly before the junction with the Babine River. If we take 345 the location of the station GRDC No. 4245920 on 30', we get the Skeena and the Babine River joined together. We have to move the station to allocate it to the correct basin. The reported UPA of the station is 12,400 $\text{km}^2$. If we had selected only by upstream area or by weighted upstream area and distance, we would have chosen the Babine River (UPA of 30': 12,495 $\text{km}^2$) in preference to the Skeena River (UPA of 30': 11,937 $\text{km}^2$). Figure 7a shows that the selected 30' basin in darker blue (Skeena River) with the lower UPA fits better with the high-resolution basin even if the distance to the cell center of the Skeena basin 350 is 59 km (distance between green dot and dark blue square) compared to the distance to the Barbine River of 28 km (distance between green dot and red square).

Figure 7b shows a station mismatch selected by the UPA at 5'. The river Khudan in Russia, GRDC No. 2907025, has a reported UPA of 7,800 $\text{km}^2$. We only shifted the station by 0.8 km to fit the 3' high-resolution network. If we select by UPA, 355 the Uda River, with an UPA on 5' of 7,901 $\text{km}^2$, fits better than the Khudana River, with an UPA on 5' of 7,673 $\text{km}^2$. Also the cell center of the Uda River is closer to the station (4.4 km) than the cell center of the Khudan River (8.2 km). Selecting by area and shape similarity points to the correct basin, shown in dark blue in Figure 7b.

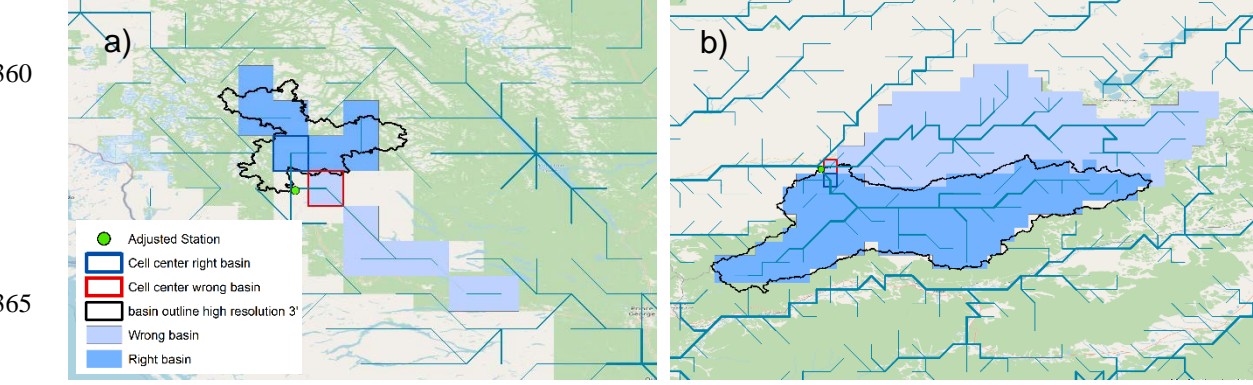

**Figure 7:** Mismatch of basin allocation because of selection from upstream area only**.** A) shows the Skeena River, Canada at 30' resolution b) shows the river Khudan, Russia at 5' resolution. © OpenStreetMap contributors 2022. Distributed under the Open Data Commons Open Database License (ODbL) v1.0.


For the coarser resolution of 5', we selected 6,414 stations with a basin area $\geq$ 1,000 km$^2$. For the resolution of 30', we selected 2,741 stations with a basin area $\geq$ 9,000 km$^2$. For the 2741 selected station resolution of 30', we found 68 cases (2%) where the station location would account for the wrong basins, which the UPA and distance method could not detect. For 684 stations (25%), we chose basin representations of the stations that fit better to similarity and UPA than to UPA and distance. For the

6414 selected stations for 5' resolution, we had 23 cases of station mismatch (0.7%) and 680 (11%) where we chose another basin representation than with UPA and distance. We assigned polygons based on the upscaled river network for those two resolutions. We provided the list of stations with adjusted station locations and the 5' and 30' watershed boundaries as shapefiles.

## 3.4 Selecting stations for calibration on low resolution 5' and 30'

Based on the selection criteria of tables 1 and 2, we included meta information of the station time series (length and end date of the time series, number of missing values, daily or monthly values). As mentioned in the method section, the selection criteria were subjective, but the Python code for changing the criteria tables is available on GitHub.

For the low resolution of 30', from the 2,724 stations (with an UPA of $\geq$ 9000 km$^2$), we selected 953 stations for calibration.

Another 105 stations could be used for validation purposes. The latter stations are not in the first calibration category because they are equal to or more than 95%, similar to a station chosen for calibration. We dismissed 1,666 stations from the calibration because they do not fulfill the necessary criteria given in Table 1. For the low-resolution of 5', we selected 3,917 out of 6,415 stations. Another 175 stations were available for validation, and we dismissed 2,323 stations.

Figure 8 and the histograms in Figure 9 show the global distribution. North America, Brazil, Europe, Russia, and Australia are

well covered, but Asia and Africa only partly. On 5' resolution, 441 stations are in Africa, and 1,270 and 746 are in North America or Europe.

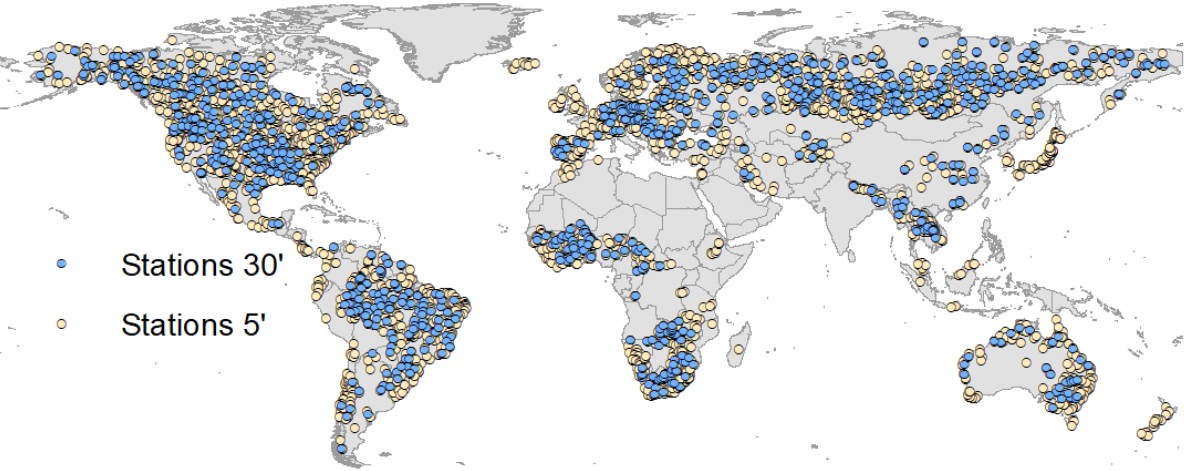

**Figure 8:** Selected station for calibration on 30' (949 stations) and 5' (3,917 stations)
(World administrative boundaries by https://www.opendatasoft.com)

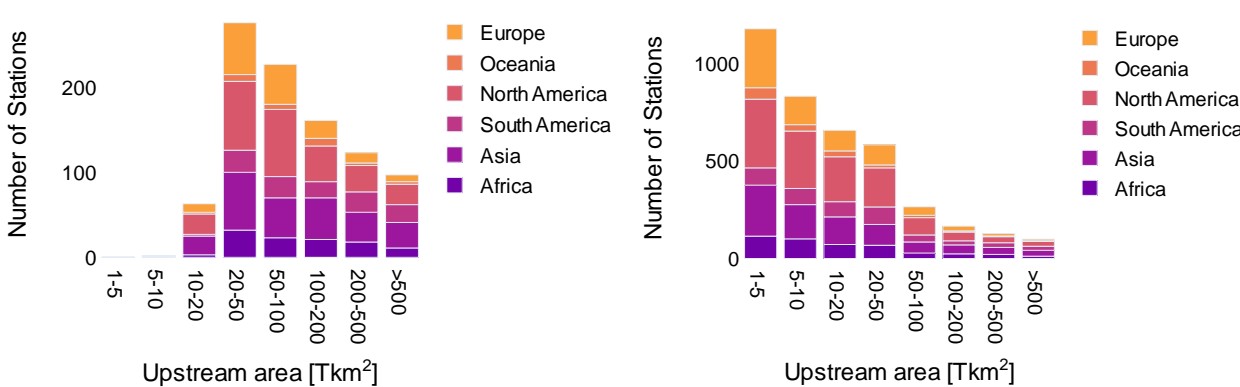

**Figure 9:** Histogram of selected calibration stations (a) 949 stations for 30' (b) 3,917 stations for 5', classified by upstream area and continent

## 4. Conclusion

This paper describes the procedure used to generate a dataset of station locations of observed discharge to be used at different resolutions for calibrating large-scale hydrological models. It is based on the metadata of GRDC stations and MERIT Hydro. The Python source code and dataset produced are freely available for download through a GitHub and Zenodo repository.


The first step toward generating a high-resolution collection of watershed shapefiles was to update the work of Lehner (2012) to include more basins (10,241 stations vs. 7,163) based on a higher resolution river network database (3'' MERIT Hydro from Yamazaki et al. (2019) vs. 15'' the HydroSHEDS from Lehner et al. (2008), including the changed GRDC IDs from September

2021. The second step, of generating a lower-resolution collection of watershed shapefiles based on the Intersection over Union ratio, was inspired by the ideas of Rezatofighi et al. (2019) and Munier and Decharme (2021). It is a better approach than selecting a station location on low-resolution river network systems based only on the UPA and distance to the original location. Here, we provide the low-resolution watershed boundaries on 30' and 5' and the source code to produce results for different resolutions and projection systems. The third step, selecting suitable stations for calibration and validation, was also based on the Intersection over Union ratio. This selection of stations can now be used more effectively to calibrate grid-based hydrological models at different resolutions.

We are very grateful for the work of GRDC in collecting and making available a considerable number of stations. Around 8,000 of the 10,701 stations fit very well and have less than a 5% UPA difference between the reported UPA and the MERIT Hydro calculated UPA. 10,000 stations have less than 30% UPA difference. For 228 stations, however, we could not find a suitable location, and for another 437 stations, the reported area and calculated area are very different (25% error). Most stations (8544) could be located on the high-resolution MERIT network within a 1 km range. However, 843 stations have a corrected station location more than 5 km distance to the original position. We propose a quality check for these stations; otherwise, the time series cannot be used for any application.

## 5. Code and data availability

The MERIT Hydro - global hydrography dataset is available for download at http://hydro.iis.u-tokyo.ac.jp/~yamadai/MERIT_Hydro (Yamazaki et al., 2019) and was last updated on 17 May 2019. The metadata information on 10,701 was provided by the Global Runoff Data Centre (GRDC, https://www.bafg. de/GRDC) (04/04/2022). The watershed boundaries based on Lehner et al. (2011) can be downloaded from GRDC.

The source code, tables, and shapefile datasets for high-resolution 3'' and low-resolution 5' and 30' are stored on Zenodo with the associated DOI https://doi.org/ 10.5281/zenodo.6906577. In addition, we provide the source code on a GitHub repository https://github.com/iiasa/CWATM_grdc_calibration_stations as release version 1.0. Here we used input data from MERIT Hydro with a resolution of 3'' and an eight-direction (D8) flow model network format, but the code can be changed to use any resolution and non-geographical projections as input format. Please keep in mind that the Zenodo repository is the location where users can retrieve exactly the data that have been used for this study.

**Acknowledgments**

The authors acknowledge the Global Runoff Data Centre (GRDC, Koblenz, Germany) for providing the metadata and time series of 10701 discharge stations. We appreciate all the other open-source projects (especially the flwdir open-source project from Eilander et al. (2021)) which we used to collect ideas and which, for our part, we hope to cross-fertilize with our ideas.

We are grateful for all the freely available data sets (especially from the MERIT Hydro database). We also thank the students of St. Pölten New Design University, class of Prof. E. Bravi, for providing advice on creating the figures. The project has received funding from European Union's Horizon EUROPE Research and Innovation Programme under Grant Agreement N° 101059264 (SOS-WATER).

**CRediT authorship contribution statement**

Peter Burek: Conceptualization, Methodology, Data curation, Writing - original draft, Writing - review and editing.

Mikhail Smilovic: Writing - review and editing.

**Competing interests**

The contact author has declared that neither they nor their co-authors have any competing interests.

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
