# Peer review of "The use of GRDC gauging stations for calibrating large-scale hydrological models"

_Earth System Science Data, 2022_

## Author Response (AR1)

**The use of GRDC gauging stations for calibrating large-scale hydrological models**

Peter Burek and Mikhail Smilovic

**Reviewer 1**

General comments

This manuscript describes the new location data of GRDC stations for calibrating large-scale hydrological models. The calibration of these models relies on the accuracy of stations allocated on the gridded river network of different spatial resolutions. The data is developed based on the idea of the 'intersection over union ratio approach', which quantifies the similarity of the watershed shapes between low- and high-resolution gridded watersheds. The new dataset's accuracy was reasonably good compared to the previous versions; thus, the dataset attracts interest in hydrological communities and is worth publishing. Although the estimated precision is promising, however, I think the manuscript does not contain an adequate description as a data paper. Given the location of GRDC stations is a widely used essential information in hydrology and earth system science studies, I think the manuscript is worth publishing on ESSD, after corrections on a few ambiguous parts.

Specific comments:

(1) L 89. Is "Upstream area error" used in the subsequent manuscript?
We used both upstream area error and upstream accordance in the paper, which might lead to confusion. We removed the term "Upstream area error" and replaced it with upstream area accordance. We also removed the term similarity index and replaced it with "Intersection over Union ratio". Both indices score low with 0 and high with 1

(2) L93-94. The unit of the distance was not described here. The values of OC calculated with the described equation differed depending on the unit of the distance used. Because the values of (1-'area accordance') are always less than one, the OC values are predominantly determined by the distance if the unit is in kilometers or meters (usually larger than 1). At least, the second term of the equation should be normalized to the range from zero to one. Also, the authors should explain how the weighting of the second term, 2, was determined.

Thanks for paying attention on this. We followed the approach of Lehner (2012), but we did not describe his method properly. The two terms are normalized to fit together, and the equations and the weighting are from Lehner (2012), too. We point this out.
We changed the description in L90f:

1. A rectangular search radius of 165 arcsec (~5 km) for each station was defined.
2. For each grid in this rectangle, the upstream drainage area (UPA) from the network from Yamazaki et al. (2019) was compared to the area reported in the GRDC, and the upstream area accordance is computed:

- o Upstream area accordance = GRDC reported UPA / gridded network UPA
  (where: GRDC reported UPA < gridded network UPA)
  Upstream area accordance = gridded network UPA / GRDC reported UPA
  (where: GRDC reported UPA ≥ gridded network UPA)
3. All cells with an upstream area accordance of less than 50% were dismissed from further evaluation.
4. A first ranking scheme – area discrepancy (RA) - was calculated with values between 0 (best fit) to 50: RA = 100 - Upstream area accordance[%]
5. For the second ranking scheme – distance (RD) – the distance of the cell to the reported station location in the GRDC database was calculated and normalized to get the value 0 at the station location and 50 in 5 km distance.
6. An objective criterion (OC) for ranking was computed by OC = RA + 2 * RD. The equation and weighting were taken from Lehner (2012).
7. The grid cell with the lowest OC value was taken as the corresponding grid cell for the station location on a high-resolution network
8. If no station location was found in this step, the search radius was increased to 5' (~10 km), OC was calculated as OC = RA + RD, and the lowest OC value was taken as the corresponding grid cell.

(3) L 141=143: The choice of the weighting factor for calculating ED should be explained.

The objective criterion Upstream area accordance range from ]0,1], and the Intersection over Union ratio criterion has a range of [0,1]. Therefore we decided to give a weighting factor of 1 to each.

We added the text:
Both objective criteria have a range between 0 and 1. Therefore, we decided to use a weighting factor of 1 for both criteria.

(4) L 144: Figures 1 (and Figure 2 as well) are not appropriate for the examples describing the automatic upscaling process based on the similarity of shape, because it seems that only the 'area accordance' suffice for the selection of station 7. I would suggest the authors select a more appropriate example. I think Figure 7 worth explains the problem of mismatch in the upscaled stations and how they are solved with the proposed procedures. Upon revising, please consider the necessity of Figure 2 as it was not referred to in the manuscript.

Yes we agree that for Passau/Inn the area accordance would be enough, but it is a good example that a) the cell with the station is not the most appropriated one b) moving 1 cell away from the station shows very different basins. We think it is fine to illustrate the method. In the result part (figure 7) we show 2 examples where Intersection over Union ratio really matters.

We changed figure 2 and added an explanation of fig 1 and 2 in the text:

"Figure 1 illustrates this method for low resolution 5' and for cell location No. 7, which is one 5' cell south of the cell where the station "Passau/Inn" is located (see the zoom in the upper left part of figure 1). Even if this cell is not representing the cell where the station is located, this cell fits the upstream area accordance and the Intersection over Union ratio best of all 25 cells around the station location."

"Figure 2 shows four examples out of the 25 cell locations around station "Passau/Inn". Figure 2a uses the cell where the station is located. This cell represents not only the Inn, but the also the Danube and the Inn basin. Figure 2b includes only a small tributary of the Inn and figure 2c contains only the Danube basin but not the Inn basin. Figure 2d shows the best location (one grid cell south of the grid cell with the station – same as in figure 1)."

**Technical corrections:**

(1) L18-21: I think this part is not relevant to the main context. Suggest deleting.
Thanks for the correction. We think you are right, and these lines can be deleted

(2) L 92: Delete ")".
Done

(3) L 133: The number of cells for each resolution does not consistent with those described in line 303. I think this line should be corrected.

We corrected the numbers in L133: "(e.g., $\geq$ 9,000 km$^2$ for 30' (~3 cells), $\geq$ 1,000 km$^2$ for 5' (~12 cells))." and in L303 - (~12 grid cells on 5').
Thanks for pointing. At the equator, a 5' grid cell has an area of 85.8 km2, and a 30' grid cell of 3087.6 km2

(4) Figure 3 was not referred to in the manuscript

we put in a description:

"Figure 3 shows the global distribution of GRDC stations (status: March 2022) with a high concentration of stations in North America and Europe and a lower and more clustered distribution in Africa and Asia."

**Reviewer 2**

The manuscript describes the procedure used to generate a dataset of station locations of observed discharge to be used at different resolutions for calibrating large-scale hydrological models. The authors update the 10-year-old database of GRDC watershed boundaries and provide source codes and high- and low-resolution watershed boundaries. The manuscript is interesting and the results are useful for scientific purposes. However, before it can be considered for publication, the authors need to undertake a thorough revision process to better explain the objectives of the manuscript and the steps needed to achieve them. In addition, figures and tables must be clearly explained in the text. Symbols and acronyms should be used consistently.

**General remarks:**

1. Authors should describe the dataset of river GRDC discharge data in the introduction. GRDC is not only a river discharge time series dataset but it also contains information on hydrometric station location, upstream basin area…In this way the readers can better follow the manuscript and in particular it is easy to understand the meaning of "*reported upstream area*" (e.g., point e in lines 57-59 and line 71).
We added in the abstract: "The Global Runoff Data Centre provides time series of observed discharges and information on hydrometric stations that are valuable for calibrating and validating the results of hydrological models."
We added in the introduction: "The GRDC database of river discharge comes which information about the stations from the data providers, like the location of the station, name of the station and the river, upstream area, elevation, mean discharge, and more. Especially the location and the upstream area are very important to compare model results from hydrological models with station discharge data."

2. The authors should better describe the objective of the manuscript, which is not only to revise and correct the shapefiles of the GRDC stations, but also to provide a Python code to easily select stations for calibration/validation of LSM models.

   We added in the abstract: "we provide source codes and high- and low-resolution watershed boundaries to easily select stations for calibration/validation of hydrological models."
   We added in the introduction: "The objective of this paper is to provide a Python code to easily select stations for calibration/validation of hydrological models by adressing these possible errors and giving examples of how to correct them."

3. The Methods section should be revised. The authors should clearly state the objectives of the study and the steps needed to achieve them. Some sentences should be added before line 75 to introduce paragraph 2.3 and its sub-paragraphs. For example, when reading line 161, it is not immediately clear what does the authors mean by "For the next selection step…".

   We added some lines before L75 to describe the step wise approach of the methods:

   "The methods can be split up into three main groups, each group building upon the results of

the previous one. The first method describes allocating a station location from the GRDC database to fit best on a high-resolution network. This method reproduces the approach from Lehner (2012). The second method describes how to upscale the station location from a high-resolution network to a low-resolution network used in standard land-surface hydrological routing models by comparing upstream area and similarity of the station upstream areas in high and low resolution. The third method describes how to select the most appropriate stations for calibrating hydrological models, depending on the metadata of the stations and the chosen model grid resolution. "

4. 84. The last part of the sentence "or with no upstream area record" is misleading. If I understand correctly, this part should be deleted as in the paragraph 2.1.1 only stations with an upstream drainage area are considered.
We added some numbers in L84f and added reported upstream area.
"For the evaluation, we used all stations with an reported upstream area greater than or equal to 10 km$^2$ (124 stations have an upstream area smaller than 10 km$^2$) or with no reported upstream area record (327 have no reported upstream area record in the GRDC dataset)."

We kept the stations with no reported upstream area in the GRDC dataset, because most of them could be clearly identified by location and most of them (201 stations) are in Africa and Asia, which are anyway underrepresented.

5. 120. Please be consistent with the notation. According to L. 44, "*30 arc minutes resolution*", should be recalled here as 30'. Please modify the sentence.
We use now 30' or 5' instead of arc minutes from L44 on. Same with 3 arc seconds.
We use 3'' after we introduced 3'' in L 72

6. 132. Please be consistent with the notation. According to L. 87, the upstream area was abbreviated as "UPA". The sentence on lines 132-133 can be changed as "*We defined a minimum UPA for the station we wanted to use in the low-resolution hydrological model (e.g., UPA ≥ 9,000 km$^2$ for 30' (~180 cells), UPA ≥ 1,000 km$^2$ for 5' (~100 cells))*."
We followed your advice and used UPA from L 87 in lines 132-133 but also everywhere else in the text and figures.

7. L 134-135. This sentence is hard to understand. Please, rephrase it.
To find the grid cell on the coarse resolution network which fits best to the upstream area and shape of the high-resolution network, we calculated two objective criteria for all coarse grid cells with a distance <= 2 coarse cell distance (altogether 25 grid cells) to the location of the station on the high-resolution network

8. 139. Please define the range of variability for the Intersection over Union ratio.
Done, it is [0,1]. We put in the text: "The Intersection over Union ratio can have a value between [0,1]. The closer to 1 the value of Intersection over Union ratio is, then the more similar the shapes are." In L 162f. We also put in the range of upstream area accordance in L 154: "The upstream area accordance can have a value between ]0,1] with 1 having GRDC and coarse area the same value." ( as ]0,1] as 0 is outside of the interval)

9. 143. What does "*similarity*" stand for? Does it refer to the Intersection over Union ratio?
Yes similarity = Intersection over Union ratio. We replaced similarity by Intersection

over Union ratio in text and figures

10. L 144. Please explain Figure 1 clearly.
We added explanation for figure 1
"Figure 1 illustrates this method for low resolution 5' and for cell location No. 7, which is one 5' cell south of the cell where the station "Passau/Inn" is located (see the zoom in the upper left part of figure 1). Even if this cell is not representing the cell where the station is located, this cell fits the upstream area accordance and the Intersection over Union ratio best of all 25 cells around the station location."

11. Table 2 should be better explained in the text.

We explained table 2:
"If a station had a higher Intersection over Union ratio or upstream area accordace than 80% it got for every 2% one scoring point. Stations earn scoring points for every five additional years of time series length and for end dates of the time series after 1985. For missing data in the time series scoring points are subtracted (see Table 2 for the scoring criteria). The station with the higher scoring points is chosen. These criteria are subjective and can be changed in the Python code"

12. L 201 vs L.107. How many stations do not have a catchment shapefile? 228 as stated in line 201 or 352 as is written in line 107? Please check.
Thanks for checking. We corrected L 107:
"For 2.2% of the stations (228 stations), we could not find an adequate location on the high-resolution network"

13. L 224. Should be Table 3 instead of Table 1. Yes, you are right, changed to table 3

14. L 225. What does "distance median" mean? Please add details.
Added:
"(Here the distance is the distance in meter between reported station location in the GRDC dataset and the location represented in the 3'' MERRIT network. The median of distance is calculated as the median of all distances in each row of table3.) "

272. Should be Lokoja station not Lokojo.
Done

15. L 316 should be Figure 7b. The caption for Figure 7 should clearly describe both the a) and b) panels.
Done L316 and we changed the caption:
Mismatch of basin allocation because of selection from upstream area only. a) shows the South Platte River, USA at 30' resolution b) shows the river Pisuerga in Spain at 5' resolution. © OpenStreetMap contributors 2022. Distributed under the Open Data Commons Open Database License (ODbL) v1.0.

16. Figure 2 is not described in the manuscript. Please explain the figure or delete it.
We added:
"Figure 2 shows four examples out of the 25 cell locations around station "Passau/Inn". Figure 2a uses the cell where the station is located. This cell represents not only the Inn, but the also the Danube and the Inn basin. Figure 2b includes only a small tributary of the Inn and figure 2c contains only the Danube basin but not the Inn basin. Figure 2d shows the best location (one grid cell south of the grid cell with the station – same as in figure 1)."

17. Figure 3 is not described in the manuscript. Please explain the figure or delete it. This figure could be moved to section 2.

We did not move this to the method section, because the method part is independent d of the actual number of stations of the GRDC database. In the result part we show the application of the methods to the GRDC database of March 2022, which include 10701 stations at that time.

We put in a description:

"Figure 3 shows the global distribution of GRDC stations (status: March 2022) with a high concentration of stations in North America and Europe and a lower and more clustered distribution in Africa and Asia."

Minor corrections:

1. 52. A parenthesis is missing at the end of the sentence. Done
2. L 53. Please, define the ISIMIP acronym.
   Deleted ISIMIP here because it does not add information here. ISIMIP is explained in section 2.2
3. 67. Please delete the parenthesis before the "3 arcseconds". Please be consistent with the conversion of 3" to metres. Here is it indicated that "*3"~93m*" whereas in Line 76 it is written "*3 arc seconds (~100 m)*". Done, we stick to 3'' *~100m, but we gave also the exact value of 3'' at the equator = 92.61 m*
4. L 67. Please, enter the corresponding metres to 15 arc seconds. Done
5. 70. Please, enter the corresponding metres to 5 arc minutes. Done
6. L 92. Please delete the parenthesis at the end of the sentence. Done

---

## Referee Report (RR1)

Burek and Smilovic proposed a new method to identify grid cells from global hydrological or river transport model mesh to match with GRDC gauges. They used the most updated GRDC gauges and delineated the corresponding basin boundaries using a 90m resolution flow direction dataset. They further allocate the GRDC gauges to two coarse resolutions meshes that used by global hydrological model. Instead of using the comparison of drainage area and distance between the identified grid cell to the gauge location, they proposed to use drainage area and basin boundary as objective. They used several basins as examples to demonstrate the improvement of their method in finding the appropriate grid cells to calibrate and validate using GRDC dataset. They further proposed some other criteria, such as measurement period, to filter the gauges that should be used.

This manuscript is a revised version, and it is my first time to review it. First, I agree with the authors that this topic is very important for large scale hydrological models. The data, especially the code will be very useful to the modelers. However, I still have some major concerns about this study.

1. How easily can the user apply the Python code to allocate GRDC gauges to their own mesh? I note there are a lot of large-scale global river network meshes that used by different models. Even at the same spatial resolution, they can have different representations of river network because different algorithms were used. Without the authors' experience, can we successfully use the Python code?

2. Theoretically, the proposed method should be more accurate than previous method. But I failed to see it from the presented results, for example, some clarifications are needed in Figure 2 and Figure 7 (see my detailed comments below). In addition, only two basins were shown to demonstrate the improvement by using the proposed method. I wonder how many gauges in total will be improved (similar to the basin in Figure 2)? This will be a critical metric to report. It will be a significant contribution If the proposed method finds improved drainage area as the previous method for a large fraction of the selected gauges.

Overall, I recommend additional revisions before publication. Please find my additional comments in the following.

Line 21: … such as Nash-Sutcliffe and Kling-Gupta .

Line 31: GSIM database (cited by the author) provides over 30,000 stations for streamflow measurement, which is more than GRDC. Also, GSIM provides the shapefile of the drainage area for each station.

Line 157: [0,1]

Figure 2: Subplot (a) seems to plot the contributing area of cell No 14 instead of cell No 12.

Line 189-Line192: This be described earlier, probably before the presentation of Figure 1 and 2.

Line 237: Do you mean use stations with UPA larger than 10km^2?

Figure 5: increase the font size. I can barely read the station number from the subplot (a). Add legend for the red and green circles.

Figure 7: What does the red circle mean? It will be helpful to plot the gauge location and allocated cell centers (both the right and wrong ones) on the map too. I don't think Figure 7b is a good example to show the benefit of the proposed method. The dark blue area is very close to the light blue area, though the light blue area is closer to UPA derived from high resolution DEM. But the outlet of the dark blue area is closer to the gauge location. So, the previous method (compare UPA and distance to original gauge location) should give us the right contributing area on the coarse resolution mesh.

Line 358: shown in dark blue in Figure 7b?

---

## Author Response (AR2)

Dear reviewer,

Thank you for reviewing our paper, your constructive comments and your attention to details. We appreciate your voluntary effort and we revised the manuscript according to your comments. The comments have been addressed as following:

How easily can the user apply the Python code to allocate GRDC gauges to their own mesh? I note there are a lot of large-scale global river network meshes that used by different models. Even at the same spatial resolution, they can have different representations of river network because different algorithms were used. Without the authors' experience, can we successfully use the Python code?

We already have some feedback from users on implementing our Python code. Some Python expertise is needed to adapt the code to your needs. We have a documentation on GitHub https://github.com/iiasa/CWATM_grdc_calibration_stations and zenodo https://doi.org/ 10.5281/zenodo.6906577 where each module, the input and output files are described. Even if models use a different representation of river network there is a kind of standard format to describe the eight-*direction* (D8) *flow* model network. We are using the D8 direction codings also used by the MERIT Hydro Dataset (Yamazaki et al., 2019), which is also used in ArcGIS https://pro.arcgis.com/en/pro-app/latest/tool-reference/spatial-analyst/how-flow-direction-works.htm. In case you use another direction coding like the LDD network https://pcraster.geo.uu.nl/pcraster/4.4.0/documentation/pcraster_manual/sphinx/secdatbase.html you can adapt our code quite easily or using the pyflwdir library https://pypi.org/project/pyflwdir/0.4.3/ to convert formats from Eilander et al. 2021, which has a good documentation on flow direction. In addition, we are not limited to 5 or 30 arcmin. Changing our script to other resolutions and non lat/lon projections is also possible.

We added to the paper in 5. Code and data availability Line 432ff:
We used input data from MERIT Hydro with a resolution of 3'' and an eight-*direction* (D8) *flow* model network format, but the code can be changed to use any resolution and non-geographical projections as input/output format.

Theoretically, the proposed method should be more accurate than previous method. But I failed to see it from the presented results, for example, some clarifications are needed in Figure 2 and Figure 7 (see my detailed comments below). In addition, only two basins were shown to demonstrate the improvement by using the proposed method. I wonder how many gauges in total will be improved (similar to the basin in Figure 2)? This will be a critical metric to report. It will be a significant contribution If the proposed method finds improved drainage area as the previous method for a large fraction of the selected gauges.
Thank you, for this question. This number would indeed show the advantage of the proposed method.
We changed line 374ff:
For the 2741 selected station resolution of 30', we found 68 cases (2%) where the station location would account for the wrong basins, which the UPA and distance method could not detect. For 684 stations (25%), we chose basin representations of the stations that fit better to similarity and UPA than to UPA and distance. For the 6414 selected stations for 5' resolution, we had 23 cases of station mismatch (0.7%) and 680 (11%) where we chose another basin representation than with UPA and distance.

Line 21: … such as Nash-Sutcliffe and Kling-Gupta for calibrating global hydrological models.
We changed this

Line 31: GSIM database (cited by the author) provides over 30,000 stations for streamflow measurement, which is more than GRDC. Also, GSIM provides the shapefile of the drainage area for each station.
We added this sentence above.

Line 157: [0,1]

Figure 2: Subplot (a) seems to plot the contributing area of cell No 14 instead of cell No 12.
Thank you for looking so carefully at this. Yes indeed, it is cell No 14.

Line 189-Line192: This be described earlier, probably before the presentation of Figure 1 and 2.
We put it before Figure 1

Line 237: Do you mean use stations with UPA larger than 10km^2?
We changed to:  an UPA larger than or equal to 10 $km^2$

Figure 5: increase the font size. I can barely read the station number from the subplot (a). Add legend for the red and green circles.
Changed figures and increased the font size

Figure 7: What does the red circle mean? It will be helpful to plot the gauge location and allocated cell centers (both the right and wrong ones) on the map too. I don't think Figure 7b is a good example to show the benefit of the proposed method. The dark blue area is very close to the light blue area, though the light blue area is closer to UPA derived from high resolution DEM. But the dark blue area outlet is closer to the gauge location. So, the previous method (compare UPA and distance to original gauge location) should give us the right contributing area on the coarse resolution mesh.
Thank you for pointing this out. We didn't check with the UPA and distance for comparison.
Even for 7a you might find the right basin with the UPA and distance method, because the right basin is 107 km from the station point and the wrong one 120 km.
Therefore, we choose better examples where only the similarity method gets to the right result.
We changed line 344ff:
The station "Above Babine River" of the Skeena River in Canada, GRDC No. 4245920, is the station shortly before the junction with the Babine River. If we take the location of the station GRDC No. 4245920 on 30', we get the Skeena and the Babine River joined together. We have to move the station to allocate it to the correct basin. The reported UPA of the station is 12,400 $km^2$. If we had selected only by upstream area or by weighted upstream area and distance, we would have chosen the Babine River (UPA of 30': 12,495  $km^2$) in preference to the Skeena River (UPA of 30': 11,937 $km^2$). Figure 7a shows that the selected 30' basin in darker blue (Skeena River) with the lower UPA fits better with the high-resolution basin even if the distance to the cell center of the Skeena basin is 59 km (distance between green dot and dark blue square) compared to the distance to the Barbine River of 28 km (distance between green dot and red square).
Figure 7b shows a station mismatch selected by the UPA at 5'. The river Khudan in  Russia, GRDC No. 2907025, has a reported UPA of 7,800 $km^2$.  We only shifted the station by 0.8 km to fit the 3' high-resolution network. If we select by UPA, the Uda River, with an UPA on 5' of 7,901 $km^2$, fits better than the Khudana River, with an UPA on 5' of 7,673 $km^2$. Also, the cell center of the Uda River is closer to the station (4.4 km) than the cell center of the Khudan River (8.2 km). Selecting by area and shape similarity points to the correct basin, shown in dark blue in Figure 7b.

Line 358: shown in dark blue in Figure 7b?
Corrected

In addition, we added in acknowledgment:
The project has received funding from European Union's Horizon EUROPE Research and Innovation Programme under Grant Agreement N° 101059264 (SOS-WATER).